# Targeted Protein Degradation Systems: Controlling Protein Stability Using E3 Ubiquitin Ligases in Eukaryotic Species

**DOI:** 10.3390/cells13020175

**Published:** 2024-01-17

**Authors:** Yoshitaka Ogawa, Taisei P. Ueda, Keisuke Obara, Kohei Nishimura, Takumi Kamura

**Affiliations:** Department of Biological Science, Division of Natural Science, Graduate School of Science, Nagoya University, Nagoya 464-8601, Japan; ogawa.yoshitaka.e7@s.mail.nagoya-u.ac.jp (Y.O.); ueda.taisei.c9@s.mail.nagoya-u.ac.jp (T.P.U.); obara.keisuke.r2@f.mail.nagoya-u.ac.jp (K.O.)

**Keywords:** target protein degradation, ubiquitin, ubiquitin proteasome pathway, E3 ubiquitin ligase, PROTACs

## Abstract

This review explores various methods for modulating protein stability to achieve target protein degradation, which is a crucial aspect in the study of biological processes and drug design. Thirty years have passed since the introduction of heat-inducible degron cells utilizing the N-end rule, and methods for controlling protein stability using the ubiquitin–proteasome system have moved from academia to industry. This review covers protein stability control methods, from the early days to recent advancements, and discusses the evolution of techniques in this field. This review also addresses the challenges and future directions of protein stability control techniques by tracing their development from the inception of protein stability control methods to the present day.

## 1. Introduction

Techniques that target genomic functions at the DNA or mRNA level are powerful tools for disrupting the functions of proteins encoded by specific genes. The Cre/lox system has been widely used to disrupt target genes [1,2]. Recently, transcription activator-like effector nucleases (TALENs) [3] and Crispr/Cas9 systems [4,5] are frequently used for target gene disruption. Conversely, the tet/dox system is used to regulate the transcription of target genes [6]. At the RNA level, antisense oligos [7] and RNA interference [8,9] have also been adopted as techniques to achieve post-transcriptional gene silencing.

In the field of molecular biology and drug development, researchers often focus on directly targeting specific proteins to modulate their functions. This is typically accomplished through the use of inhibitors and activators. These compounds not only serve in biological experiments, both in vivo and in vitro, for the purpose of perturbing the function of specific proteins but are also employed as therapeutic agents for treating particular diseases. To explore potential therapeutic drugs for various diseases, pharmaceutical companies have conducted drug screening against various disease-related proteins. However, these drugs face limitations in inhibiting or activating the functions of proteins lacking enzymatic activity, termed “undruggable proteins”, as they primarily target the enzymatic reactions. Approximately 85% of the human proteome is estimated to consist of these undruggable proteins [10]. One innovative approach to inhibit the functions of these undruggable proteins involves directly degrading the target proteins. Recently, techniques that directly regulate target protein stability have been adopted to disrupt the functions of target proteins.

Protein stability affects various biological processes, including cellular responses to stress, differentiation, and cell proliferation. Protein stability is regulated through various factors, such as temperature, binding partners, and external stress through protein degradation in cells. Eukaryotic cells have two types of protein degradation systems: highly selective ubiquitin-dependent protein degradation (the ubiquitin–proteasome pathway) and autophagy for low-selectivity and large-scale degradation. Various target protein degradation techniques have been developed in recent years using these protein degradation systems. These techniques have been used as protein-knockdown tools to study biological events. Recently, they have received considerable attention in the drug design field.

Historically, temperature-sensitive mutants (ts mutants) have primarily been related to protein stability in budding yeast, which is one of the most popular conditional mutants in eukaryotic cells [11]. At restricted temperatures, ts mutations can affect the structure and/or function of the protein in a way that often makes it unstable and induces degradation, even though the protein is stable at permissive temperatures, allowing for normal cellular processes to occur. This shift to a restricted temperature leads to various cellular defects in mutant cells. These ts mutants are useful for investigating the physiological functions of essential proteins related to important cellular events, such as the cell cycle, DNA replication, and DNA segregation [12,13,14].

Many valuable mutants have been isolated for forward genetic screening. However, decoding the yeast genome makes it possible to artificially generate conditional mutants in a reverse genetic manner in budding yeast. Some conditional mutants have been generated using the heat-inducible degron system based on the N-end rule, in which protein stability depends on the N-terminal residue [15,16]. In this system, a target protein fused with an N-degron tag at its N-terminus is rapidly degraded at a restrictive temperature. The N-degron tag consists of at least two elements: a destabilizing N-terminal residue and a specific Lys residue for multiple ubiquitination. Therefore, the target protein is degraded via the ubiquitin–proteasome pathway. These heat-inducible degron mutants have been successfully used to understand the functions of essential proteins in budding yeast [17,18].

The generation of conditional mutants in a reverse genetic manner has been further attempted for application in other eukaryotic species [19,20]. However, these attempts have problems to be resolved because of unexpected side effects caused by a dynamic temperature shift. In this review, we highlight recent advances in versatile methods for modulating protein stability to degrade target proteins.

## 2. Methods for Regulating Protein Stability via Small Molecules

In the 2000s, two methods were developed to control the stability of target proteins using small chemical compounds without relying on dynamic temperature shifts. Here, we present pioneering approaches for regulating target protein stability, namely the Sheild-1 and the auxin-inducible degron (AID) systems. Both methods enable reversible control of the target protein’s stability through the addition or removal of chemical compounds. However, in the AID system, the addition of auxins triggers the degradation of the target protein, whereas in Sheild-1, its addition results in the stabilization of the target protein.

### 2.1. Shield-1

Rapamycin is a well-known antibiotic that inhibits mTOR kinase [21]. Rapamycin is also known to promote tight heterodimer formation between the FK506-binding protein (FKBP) and the FKBP-rapamycin-binding domain of the FKBP12-rapamycin-associated protein (FRB), which has been used to control the subcellular localization of a target protein to perturb its function in a rapamycin-dependent manner [22]. As rapamycin binds to endogenous mTOR and inhibits its function, rapamycin analogs that do not bind to or inhibit the endogenous mTOR complex have been developed using bump-and-hole techniques. In this process, a rapamycin analog, C20-methyllyl-rapamycin (MaRap), which binds to mutated FRB but not to wild-type FRB, was developed [23]. Interestingly, the proteins that fused with this mutated FRB became unstable in the absence of MaRap, but the levels of the mutated FRB fusion proteins were rescued with the addition of MaRap.

However, MaRap has two problems. First, two proteins (FRB and FKBP) are required for MaRap’s stabilization. Second, MaRap is difficult and expensive to synthesize. Banaszynski et al. overcame these problems by developing a “single domain and single ligand” system that controls protein stability via a small molecule, Shield-1. In the system using Shield-1, the protein fused with a destabilization domain is destabilized in the absence of Shield-1, and this destabilization is relieved by Shield-1 (Figure 1A) [24,25].

### 2.2. The Auxin-Inducible Degron System

Since its discovery more than 90 years ago, the phytohormone auxin indole-3-acetic acid (IAA) has been implicated in every aspect of plant growth and development [26]. In plant cells, auxins control transcription, which involves two families of proteins: auxin response factors (ARFs) and Aux/IAA proteins [27]. ARFs directly bind to DNA and function as transcription factors, whereas Aux/IAA proteins bind to ARFs through a conserved dimerization domain and inhibit ARFs [28].

Auxins promote the interaction between the Aux/IAA proteins and the auxin receptor TIR1, a component of the E3 ubiquitin ligase SCF complex [29,30]. Most Aux/IAA proteins have four conserved domains (I–IV). Among these, mutations in domain II stabilize Aux/IAA proteins, resulting in defects in the auxin response [31]. This indicates that domain II of Aux/IAA acts as a degron recognized by SCF^TIR1^. In general, phosphorylation is required for substrate recognition by E3 ubiquitin ligases. In contrast, auxin-mediated ubiquitination does not require phosphorylation. Auxins facilitate direct binding between TIR1 and its substrates, AUX/IAA proteins, acting as a molecular glue in plant cells [32].

TIR1, Aux/IAA, and auxins are unique factors involved in auxin-dependent degradation. Other components of the SCF complex, as well as the ubiquitin–proteasome pathway, are conserved across eukaryotic species. A target protein degradation system known as the auxin-inducible degron (AID) system was developed by transplanting these factors into non-plant eukaryotes [33]. Initially, TIR1 and IAA17 from *Arabidopsis thaliana* were used to develop an AID system. This system worked in budding yeast cultured at 25 °C, but not in vertebrate cells cultured at 37 °C. A vertebrate AID system was successfully developed by replacing *A. thaliana* TIR1 with TIR1 derived from *Oryza sativa* (OsTIR1) (Figure 1B) [33,34].

### 2.3. Improved AID Systems

Conventional AID systems are widely used across several eukaryotic species. However, the main problem of this system is the requirement of more than 100 µM of auxins for target protein degradation, which may cause cytotoxicity. Moreover, in some cases, the target proteins are severely degraded, even in the absence of auxins, which is known as “basal degradation” possibly due to endogenous auxin-like compounds [35,36]. Therefore, improved AID systems have been developed to address these problems. Sathyan et al. developed an ARF–AID system, in which basal degradation is reduced by additionally expressing the ARF16-PB protein in the AID system [37]. Li et al. developed an efficient AID system with the *A. thaliana* TIR1 homologs AFB2 and IAA7 instead of OsTIR1 and IAA17, respectively [38].

Another approach is based on the bump-and-hole technique to optimize the interaction between auxins and TIR1. Pairs of synthetic auxins and TIR1 mutants have been generated to hijack the plant auxin response system [39]. These TIR1 mutants are strongly activated by synthetic auxins rather than by the natural auxin IAA. Among them, the F79G mutation in AtTIR1 was less activated by natural auxins. An AID2 system was developed using the pair of OsTIR1F74G and a synthetic auxin, 5-phenyl-IAA [40,41]. The F79A mutation in AtTIR1 is most sensitive to the synthetic auxin 5-adamantyl-IAA [42]. The super-sensitive AID (ssAID) system was developed using the pair of OsTIR1F74A and 5-adamantyl-IAA (Figure 1C) [43]. In these new AID systems, fewer auxins are required for target protein degradation, eliminating the cytotoxicity caused by excess auxins. Furthermore, these systems can overcome basal degradation.

## 3. Target Protein Degradation Systems Derived from Proteolysis-Targeting Chimeras

Proteolysis-targeting chimeras (PROTACs) have received a lot of attention in the field of drug design because they can target proteins previously considered “undruggable” for degradation. PROTACs are heterobifunctional chemical compounds comprising three components: a target-protein-binding moiety, a linker, and an E3 ubiquitin ligase moiety. These chemicals promote the interaction between target proteins and E3 ubiquitin ligases, resulting in target protein degradation via the ubiquitin–proteasome pathway (Figure 2A) [44,45]. Several protein degradation systems have been developed based on this technique. In these systems, the protein region bound to PROTACs is used as a degron tag for degradation. The target protein fused to the degron tag is degraded in a PROTAC-dependent manner.

### 3.1. The dTAG System

dFKBP-1 promotes the integration of the FKBP12 protein with cereblon (CRBN), a component of the RING-type E3 ubiquitin ligase. In the presence of dFKBP-1, FKBP12 binds to CRBN, resulting in its degradation via the ubiquitin–proteasome pathway. dFKBP-1 analogs (dTAGs) that do not affect endogenous FKBP12 have been isolated using the bump-and-hole technique. Among the isolated dTAGs, dTAG13 was found to selectively bind to FKBP12^F36V^ but not to endogenous FKBP12. In the dTAG system, the target protein fused with FKBP12^F36V^ is degraded in a dTAG13-dependent manner without degrading endogenous FKBP12 (Figure 2B) [46].

### 3.2. HaloPROTAC

The HaloTag is a 297-residue protein with a molecular weight of approximately 33 kDa. It has a modified dehalogenase designed to covalently bind to synthetic ligands, such as fluorescent dyes and biotin, in living cells. HaloPROTAC is a chemical ligand that promotes the interaction between HaloTag and the von Hippel–Lindau (VHL) tumor suppressor, a component of the RING-type E3 ubiquitin ligase, which is also used as a target of some PROTACs [47,48]. The HaloTag-fused protein is degraded in a HaloPROTAC-dependent manner (Figure 2C). Initially, HaloPROTAC3 was used as a ligand for degradation. HaloPROTAC (HaloPROTAC-E) increased the degradation efficiency of target proteins, with 95% of the target protein degraded by 300 nM HaloPROTAC-E [49].

### 3.3. NanoTAC

NanoLuc is a minimal luciferase derived from *Oplophorus gracilirostris*. NanoLuc fusion proteins can be used in luminescence assays with the addition of substrates. In the NanoTAC system, NanoLuc is used as a degron tag [50]. NanoTAC4 was developed as a PROTAC that promotes the interaction between NanoLuc and CRBN and induces the degradation of the target protein fused with NanoLuc in the ubiquitin–proteasome pathway (Figure 2D).

## 4. Protein Degradation Systems Using Single-Domain Antibodies (sdAbs)

In the ubiquitin–proteasome pathway, substrates are recognized by the substrate recognition domain of E3 ubiquitin ligases. PROTACs are small chemical compounds that promote interactions between substrates and E3 ubiquitin ligases. In other words, the target protein undergoes ubiquitination by binding to the E3 ubiquitin ligase via PROTACs. In protein-based PROTACs methods, the interaction between the target protein and the E3 ubiquitin ligase is promoted by chimeric proteins containing sdAbs, such as nanobodies [51,52]. As a result, these chimeric proteins act like PROTACs and induce target protein degradation.

### 4.1. deGradFP

deGradFP (degrade green fluorescent protein) is a pioneering protein-based targeted protein degradation (TPD) system that was developed as a tool for understanding protein function in cells [53,54]. Caussinus et al. produced a chimeric protein comprising an E3 ubiquitin ligase component and an anti-GFP nanobody, vhhGFP4. They produced Slmb-vhhGFP4 by fusing the F-box domain of a *Drosophila* F-box protein, Slmb, with vhhGFP4, and successfully degraded GFP fusion proteins in an Slmb-vhhGFP expression-dependent manner in *Drosophila* embryos and human HeLa S3 cells.

### 4.2. The Affinity-Directed Protein Missile System

The affinity-directed protein missile (AdPROM) system is a TPD system that utilizes VHL, a substrate recognition subunit of EloBC-Cul2-type E3 ubiquitin ligase [55]. Fulcher et al. produced a VHL-aGFP chimeric protein, in which an anti-GFP nanobody, aGFP, was fused to the full-length VHL protein at its C-terminus. They found degradation of the GFP fusion protein in U2OS and HEK293 cells infected with a retrovirus for VHL-aGFP expression. In addition, they replaced aGFP with a monobody that recognized the SHP2 protein or with a nanobody that recognized the ASC protein. As a result, endogenous SHP2 and ASC proteins were successfully degraded in cultured human cell lines.

### 4.3. Antibody RING-Mediated Destruction

Antibody RING-mediated destruction (ARMeD) is a TPD system using RNF4, a RING-type E3 ubiquitin ligase [56]. Ibrahim et al. replaced the substrate recognition domain of RNF4 with an anti-GFP nanobody (GNB) to produce a GNB-RING chimeric protein. When HeLa Flp-in/T Rex cells were constructed, in which the GNB-RING protein was expressed in a doxycycline-dependent manner, degradation of YFP fusion proteins was observed in this cell line. Furthermore, they replaced GNB with another nanobody that recognized NEDP1 and successfully degraded endogenous NEDO1. They also showed that the introduction of purified chimeric proteins into HEK293T cells through electroporation induced endogenous NEDP1 degradation within 30 min.

### 4.4. bioPROTACs

Lim et al. developed chimeric proteins in which the substrate recognition domain was replaced with the binding domains of target proteins and named them bioPROTACs (Figure 3A) [57]. They systematically tested the degradation efficiency of green fluorescent protein (GFP) fusion proteins using chimeric proteins composed of an E3 ubiquitin ligase and GFP binders in HEK293 cells. They chose vhhGFP4-SPOP as a model of bioPROTACs and evaluated the efficiency of the GFP fusion protein by replacing vhhGFP4 with other GFP binders, including a nanobody, DARPin, αRep, and a monobody. These GFP binders efficiently induced GFP fusion proteins. Furthermore, seven of the ten different types of E3 ubiquitin ligases were used to generate bioPROTACs that could efficiently degrade GFP fusion proteins. They also found that when using Con1-SPOP, in which SPOP is fused with the Con1 peptide, a PCNA-binding motif protein could degrade endogenous PCNA efficiently. These results suggest the highly flexible selectivity of E3 ubiquitin ligases and the binding domains of target proteins in bioPROTACs.

### 4.5. The Trim-Away System

Antibody-bound pathogens in the trim-away system are recognized by cytosolic receptors. TRIM21 is an E3 ubiquitin ligase that binds to the Fc domains of IgG antibodies. During infection, TRIM21 recruits antibody-bound pathogens to the proteasome for degradation via the ubiquitin–proteasome pathway. In the trim-away system, two steps are necessary: first, the introduction of an antibody against the target protein, and second, the introduction of exogenous TRIM21 expression if the expression level of endogenous TRIM21 is not enough for degradation. Clift et al. successfully degraded GFP in mammalian cells, including primary cells, using an antibody against GFP (Figure 3B) [58,59]. Additionally, they produced an anti-GFP nanobody–Fc domain fusion protein. In mouse oocytes injected with mRNA encoding this fusion protein, H2B-GFP was degraded. They also succeeded in degrading endogenous Eg5 proteins using an antibody recognizing the Eg5 protein and in degrading the target protein through microinjections of the antibody.

### 4.6. The Affinity Linker-Based Super-Sensitive AID System

To reduce the cytotoxicity caused by high IAA concentrations and basal degradation, improved AID systems, including ssAID and AID2, have been developed in vertebrate cell lines and fission yeasts [40,43]. In all of these AID systems, fusion with the AID tag is necessary for target protein degradation. As fusion of the AID tag might affect the function or stability of the target protein, an AID system that does not require AID tag fusion was developed. Daniel et al. applied the anti-GFP nanobody VHHGFP4 to a conventional AID system [60]. They produced a mAID-tagged VHHGFP4 fusion protein to degrade GFP fusion proteins. To avoid the degradation of this fusion protein, lysine residues, which are the ubiquitination sites of the protein, were replaced with arginine. These authors successfully degraded GFP fusion proteins in zebrafish and human cell lines.

Ogawa et al. applied nanobodies to a ssAID system to develop an affinity linker-based super-sensitive AID (AlissAID) system in budding yeast (Figure 3C) [61]. They used the anti-GFP nanobody, VHHGFP4, to degrade the GFP fusion protein, and an anti-mCherry nanobody, LaM2, to degrade mCherry fusion proteins. Of note, they tested three different nanobodies towards mCherry and found that the degradation efficiency of mCherry-fused proteins differed depending on the nanobody. This result indicated that the proportion of antibodies was strictly affected by the degradation of the target protein in this type of degradation system.

## 5. Discussion and Future Perspectives

Various techniques have been developed to control the expression of target proteins at the DNA, RNA, and protein levels. Artificial control of target protein stability has been used to generate conditional mutants in eukaryotic cells. Most of these techniques utilize E3 ubiquitin ligases and induce target protein degradation under certain conditions, such as temperature shifts, the presence of chemical compounds, or chimeric protein introduction. During the early stages of the development of these techniques, the stability control mechanism of proteins inherent to organisms was used for target protein degradation. In recent years, with advancements in protein engineering and chemical biology, small chemical molecules involved in protein stability have been explored, and improvements in these techniques have been achieved by enhancing the interactions between these chemical compounds and the target proteins. Specifically, in the case of PROTACs, it is possible to artificially design chemical compounds that degrade target proteins by linking compounds binding to E3 ubiquitin ligases with compounds binding to the target protein.

TPD systems using E3 ubiquitin ligases have mainly used the Cullin-RING-type E3 ubiquitin ligase. More than 600 E3 ubiquitin ligases have been identified in humans [62]. It is not yet fully understood which E3 ubiquitin ligase is the most suitable. The selection of an appropriate E3 ubiquitin ligase is crucial for TPD, and different E3 ligases may be optimal for different proteins or under different conditions, such as in different cell types or tissues. Additionally, various factors are involved in the ubiquitin–proteasome pathway, including p97. The optimization of these factors may have a positive effect on the efficiency of TPD. There are also protein degradation systems that utilize another pathway. While the ubiquitin–proteasome system plays a role in selective protein degradation, mainly in the cytoplasm, autophagy is a major degradation mechanism that breaks down the cellular components of lysosomes. An autophagy-mediated degradation system can be used to target much larger cellular components than those degraded in the process facilitated by ubiquitin.

In recent years, protein-based TPD systems have been developed that use protein binders, such as antibodies, instead of chemical compounds. Antibodies bind specifically to substances, making them important tools in molecular biology. Furthermore, owing to their ability to specifically bind to target proteins, these antibodies are employed as recognition domains for TPD. In these techniques, the substrate recognition domain of E3 ubiquitin ligases is replaced with a single-domain antibody, including a nanobody. Various types of single-peptide antibodies have been developed, each possessing different properties, such as stability and functionality, in cells. However, the type of antibody that is most appropriate for TPD in cells is not yet fully understood. In other words, the properties of antibodies that are important for the efficiency of TPD are unknown. Antibody molecules can recognize subtle differences, such as single amino acid mutations or protein modifications, such as phosphorylation. In the future, the use of highly specific antibodies may enable the degradation of specific target proteins with subtle differences.

Target protein degradation systems using the ubiquitin–proteasome pathway have originated from basic research and are progressing towards clinical implementation. These innovative approaches are noteworthy for their ability to target conventional “undruggable proteins”. However, each of these systems presents its unique set of challenges, posing difficulties in their widespread clinical applications. On the other hand, new technologies are continuously emerging to support these techniques. Advancements in screening techniques for single-peptide antibodies, genome editing, and drug design are actively being developed. These technical advances, in turn, are giving rise to novel systems. In the future, it is anticipated that a more versatile target protein degradation system will emerge through research that integrates and encompasses all of these diverse systems.

## Figures and Tables

**Figure 1 cells-13-00175-f001:**
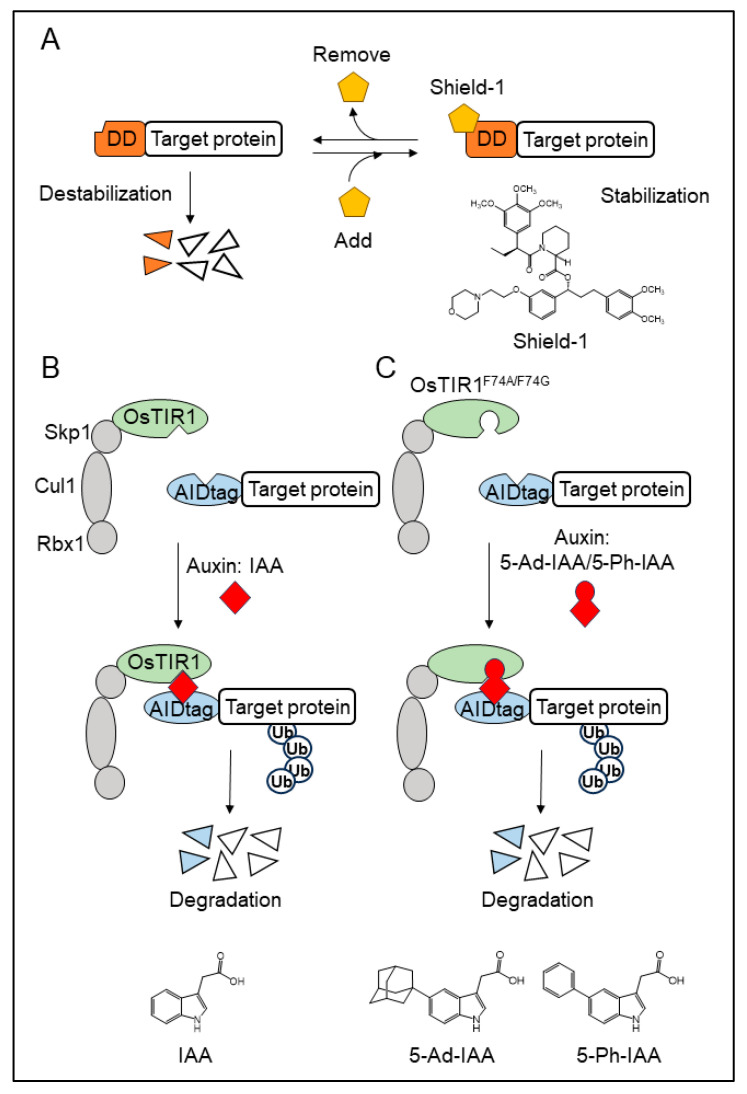
Methods for regulating target protein stability via small molecules. (**A**) A schematic illustration of the method for regulating target protein stability using Shield-1. The target protein that fused with a destabilization domain (DD) becomes unstable in the absence of Shield-1, but becomes stable in the presence of Shield-1. (**B**) A schematic illustration of the auxin-inducible degron (AID) system. The AID-tagged target protein is recruited to SCF^OsTIR1^ in the presence of the natural auxin, IAA, and degraded by 26S proteasomes. (**C**) A schematic illustration of improved AID (ssAID and AID2) systems. The AID-tagged target protein is recruited into the SCF complex, which contains a mutated OsTIR1 and OsTIR1F74A for ssAID and OsTIR1F74G for AID2. The target proteins are ubiquitinated and degraded in the presence of synthetic auxins (5-Ad-IAA for ssAID and 5-Ph-IAA for AID2, respectively).

**Figure 2 cells-13-00175-f002:**
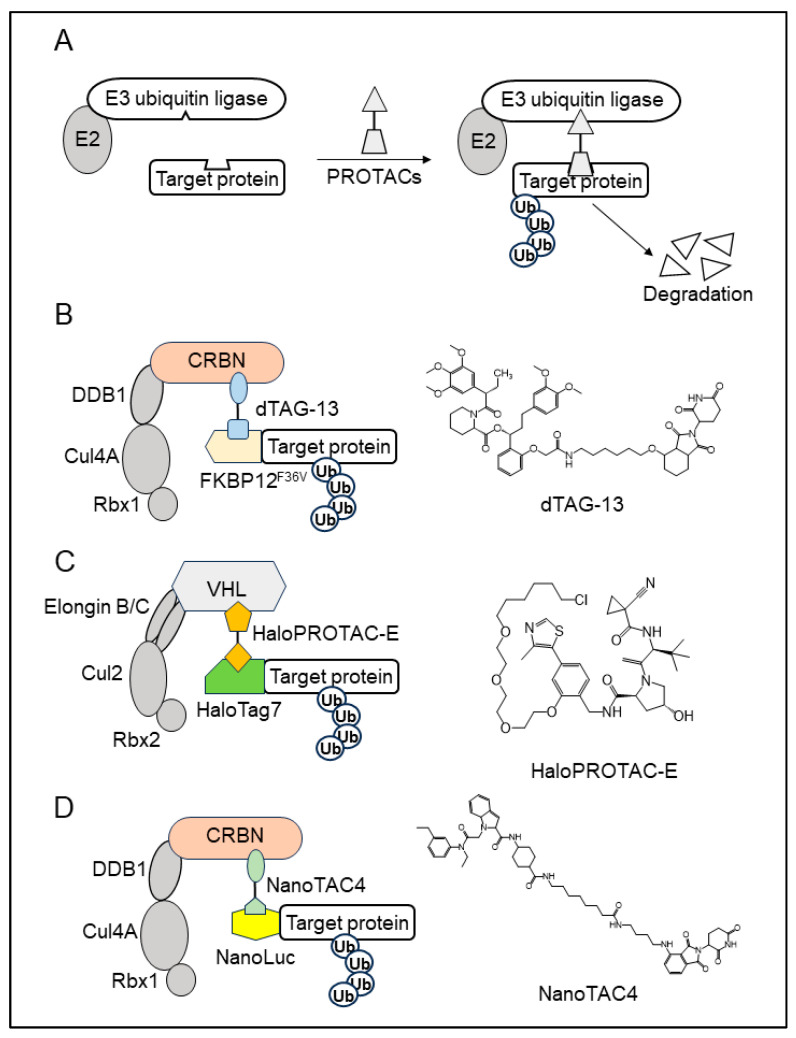
Target protein degradation system derived from PROTACs. (**A**) A schematic illustration of proteolysis-targeting chimeras (PROTACs). PROTACs are chemical compounds that bind to both E3 ubiquitin ligases and the target protein. PROTACs promote the interaction between E3 ubiquitin ligases with target proteins and induce target protein degradation. (**B**) A schematic illustration of the dFKBP-1 analog (dTAG) system. dTAG-13 is a chemical compound that promotes the interaction between FKBP12F36V and CRBN, which is a component of the Cul4-type E3 ubiquitin ligase. The target protein fused with FKBP12F36V is degraded in the presence of dTAG-13 via the ubiquitin–proteasome pathway. (**C**) A schematic illustration of HaloPROTACs. HaloPROTAC-E is a chemical compound that promotes the interaction between HaloTag7 and VHL, which is a component of the Cul2-type E3 ubiquitin ligase. The HaloTag7-fused target protein is degraded in the presence of HaloPROTAC-E. (**D**) A schematic illustration of NanoTAC. NanoTAC is a chemical compound that promotes the interaction between CRBN and NanoLuc, which is a small luciferase protein. The NanoLuc-fused target protein is degraded in the presence of NanoTAC.

**Figure 3 cells-13-00175-f003:**
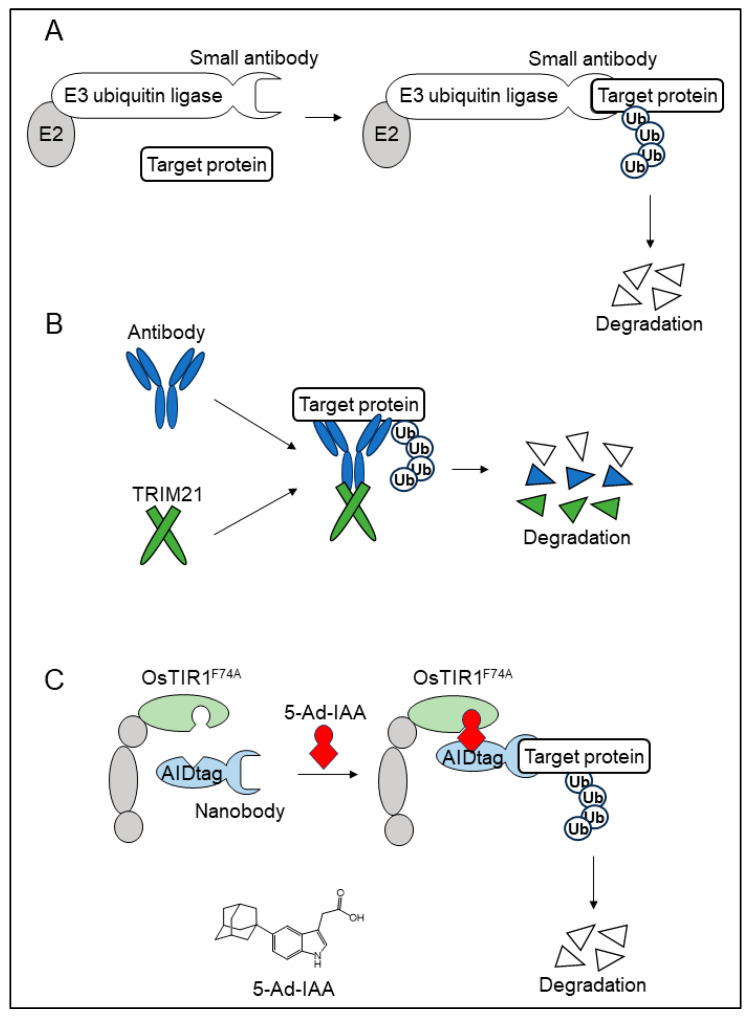
Target protein degradation systems using single-peptide antibodies. (**A**) The concept of bioPROTACs. BioPROTACs are fusion proteins of a component of an E3 ubiquitin ligase with a small-peptide binder, such as an antibody. The target protein recognized by the small-peptide binder is degraded via the ubiquitin–proteasome pathway. (**B**) A schematic illustration of the trim-away system. In cells, the target protein bound to an antibody is ubiquitinated in a Trim21-dependent manner. In the trim-away system, after the introduction of Trim21 and a specific antibody against the target protein, the target protein is degraded via the ubiquitin–proteasome pathway. (**C**) A schematic illustration of the AlissAID system. In the AlissAID system, an AID-tagged single-domain antibody (sdAb) is co-expressed with OsTIR1F74A. The target protein that binds to the sdAb is recruited to the SCF complex in a 5-Ad-IAA-dependent manner and degraded via the ubiquitin–proteasome pathway.

## Data Availability

Not applicable.

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
