# Peer review of "Targeted Protein Degradation Systems: Controlling Protein Stability Using E3 Ubiquitin Ligases in Eukaryotic Species"

_cells, 2024, doi:10.3390/cells13020175_

Round 1

Reviewer 1 Report

Comments and Suggestions for Authors

Protein stability affects many cellular processes, including cell proliferation and differentiation, cell-cycle control, and the response to different stressor conditions. Irrespective of what kind of eukaryotic cells are, they share evolutionary rather well-conserved machineries to control protein stability that rely on a post-translational modification, namely ubiquitination. The ubiquitin-proteasome system (UPS) selectively degrades ubiquitin-tagged proteins, while the second one, autophagy, is involved in large-scale degradation enabling cells to massively remodel their proteome. The latter process allows cells to remove even subcellular compartments (i.e., mitochondria throughout mitophagy or peroxisomes throughout pexophagy). Not surprisingly alterations in both processes lead to disorders of various natures, including neoplasia, neurological, and related to the immune system.

By taking advantage of the experience accumulated from basic research in the field of UPS and autophagy in recent years several target protein degradation techniques have been developed.

In their review titled "Targeted Protein Degradation Systems Controlling Protein Stability Using E3 Ubiquitin Ligases in Eukaryotic Species" Ogawa Y. and coworkers historically reviewed, from the early days up to the more recent advances, different protein stability control methods and eventually discuss the recent advancements, alongside the evolution of these techniques. The issue discussed by the authors in the review is of great interest because targeted protein degradation (TPD) represents an emerging therapeutic modality. Indeed, it displays the potentiality to tackle disease-associated proteins that have been so far highly challenging to target with conventional small molecules. Overall, TPD aims to selectively target those proteins that were previously considered "undruggable".

The review is articulated into sections in which the authors first discuss the methods for regulating protein stability via small molecules (i.e., Shield-1, the plant hormone auxin, and its improved derivative AID), then that of targeted protein degradation system based on PROteolysis Targeting Chimera (PROTAC), and eventually systems for protein degradation relying on single-peptide antibody.

The manuscript provides a comprehensive overview of the field and the authors critically discuss the different methods pinpointing the advantages but highlighting also the limitations.

Overall the manuscript flows smoothly and does not present major flaws.

However, before publication, I would warmly advise the authors to check for some typos scattered throughout the manuscript (i.e., line 85, I guess Shield-1 instead of Sheid1, is it not?; lines 139 and 190: I guess the authors meant "...Systems..." instead of "...System...", is it not? since then more than a system is discussed) and to make a little bit more clear the rationale behind Section "4. Protein Degradation System using a single-peptide antibody" (lines 190-194). 

Reviewer 2 Report

Comments and Suggestions for Authors

PROTAC, or Proteolysis-Targeting Chimeras, represents a groundbreaking approach in drug development and targeted therapy. This innovative therapeutic modality is designed to degrade specific proteins within cells, offering a novel strategy for treating diseases, especially cancer. PROTACs are engineered molecules that harness the cell's natural protein degradation machinery to remove disease-causing proteins selectively. Also, important diagrams make the manuscript really informative.

After reading the manuscript I did not find any major flaws or any necessary corrections that need to be made by the authors, which could improve the manuscript. I am happy with the current version of the manuscript. 

The key aspects of PROTAC are the mechanism of action, targeted protein degradation, versatility in targeting, advantages over traditional inhibitors, and clinical application. With ongoing research and development, PROTACs hold great promise for the future of precision medicine and the treatment of various diseases. Considering this, the scientific aspect of the current review article is valuable and on demand. The authors have done a wonderful job in summarizing the key aspects, approaches, and challenges through text mining and referencing. The authors here discussed different techniques used in PROTAC with appropriate figures.

The authors describe specific challenges associated with different techniques. Also, the summary all the different techniques related to PROTAC makes the manuscripts very useful in the field. The mode of action of the techniques is demonstrated using appropriate figures, making it a useful resource among the scientific community and the readers.

I did not find major flaws or corrections that need to be made in the current version to improve the manuscript to my knowledge. However, I request authors to cross-check for any typos before submitting the final version.

The discussion and future prospective section is valid and reasonable. Specific challenges and critical aspects of the therapeutic modality are discussed in the section. This makes the review manuscript appropriate for publication.

Important published articles are cited in the reference section as per my knowledge.

Necessary figures are added to the manuscript, and I am satisfied with that.

Reviewer 3 Report

Comments and Suggestions for Authors

This review covers protein stability control methods from early to recent advancements and discusses the evolution of techniques in this field. This review also addresses the challenges and future directions of protein stability control techniques by tracing their development from the inception of protein stability control methods to the present day. It is well-written and very informative. I recommend it to be published after minor revisions.

1.      Figures 1A. The DD-Shield System combines the simplicity of using a single transgene with the dose-dependent effect of Shield-1 to stabilize the protein of interest. In other words, the application is often by adding Shield-1 to stabilize the protein of interest instead of removing it to destabilizing the protein of interest. Therefore, I suggest that the author may consider changing Figure 1A to reflect the major applications of this method.

2.      Lines 21-24: The authors may consider adding the CRISPR system and the ASO system briefly to cover other approaches.

3.      Lines 23, 25, 114, 121 …..: Change the format of the references. For example, change [1][2] to {1,2}.

4.      Line 295: The authors may consider adding a reference regarding 600 E3 ubiquitin ligases in humans. 
